# Serum Amyloid A Proteins and Their Impact on Metastasis and Immune Biology in Cancer

**DOI:** 10.3390/cancers13133179

**Published:** 2021-06-25

**Authors:** Jesse Lee, Gregory L. Beatty

**Affiliations:** 1Abramson Cancer Center, University of Pennsylvania, Philadelphia, PA 19104, USA; Jesse.Lee@pennmedicine.upenn.edu; 2Division of Hematology-Oncology, Department of Medicine, Perelman School of Medicine, University of Pennsylvania, Philadelphia, PA 19104, USA

**Keywords:** liver, immunotherapy, serum amyloid A, immune cells, metastasis, cancer, acute phase response

## Abstract

**Simple Summary:**

The liver responds to systemic inflammation and injury in a coordinated manner, called the acute phase response. While this normal physiological response aims to restore homeostasis, malignant transformation coopts this biology to increase the risk for metastasis, immune evasion, and therapeutic resistance. In this Review, we discuss the importance of acute phase response proteins in regulating cancer biology and treatment efficacy. We also consider potential strategies to intervene on acute phase biology as an approach to improve outcomes in cancer.

**Abstract:**

Cancer triggers the systemic release of inflammatory molecules that support cancer cell metastasis and immune evasion. Notably, this biology shows striking similarity to an acute phase response that is coordinated by the liver. Consistent with this, a role for the liver in defining cancer biology is becoming increasingly appreciated. Understanding the mechanisms that link acute phase biology to metastasis and immune evasion in cancer may reveal vulnerable pathways and novel therapeutic targets. Herein, we discuss a link between acute phase biology and cancer with a focus on serum amyloid A proteins and their involvement in regulating the metastatic cascade and cancer immunobiology.

## 1. Introduction

Metastasis is the primary cause of mortality in cancer [1]. Unfortunately, once cancer cells have metastasized, the standard of care most often merely stalls disease progression. As a result, there remains an unmet need for therapies capable of disrupting the metastatic process [2]. To this end, research into unraveling the complex and intertwined networks that support metastasis will be fundamental to informing the development of novel anti-metastatic strategies and for improving patient outcomes [3,4,5,6]. Here, we discuss the role of serum amyloid A1/2 (SAA) proteins in cancer. SAA are produced by hepatocytes in the setting of cancer inflammation as acute phase reactants and have been implicated in the metastatic process [7]. Moreover, SAA proteins associate with resistance to cytotoxic- and immune-based therapies, raising the possibility that intervening on SAA and their release by the liver might offer a strategy to improve outcomes for cancer patients.

## 2. Determinants of the Metastatic Cascade

The metastatic cascade is a multistep process [8]. Beginning with detachment from the basement membrane and invasion into the surrounding stroma, a cancer cell’s metastatic journey is fraught with challenges [8]. Each step in the metastatic process is increasingly difficult. As a result, metastasis has been described as largely inefficient with few disseminated tumor cells (DTC) ever successfully completing the metastatic journey [9].

Early in malignant cell transformation, DTCs intravasate into the local vasculature at the primary tumor nest. A DTC must then endure significant stress in circulation, before extravasating into a distant tissue. Once lodged within the tissue, DTCs are influenced by their surrounding microenvironment. Notably, this microenvironment can serve as a protective niche that supports DTC survival and guards against physical and immune stressors. As a result, the distant organ microenvironment is a key determinant of the fate of a DTC and its decision to proliferate, lay dormant, or die [10].

Although cancer cell-intrinsic features are important for initiating and supporting the metastatic cascade [8,11], the tropic nature of metastasis cannot be rationalized by cell-intrinsic features alone. As observed by Steven Paget in the 1890s, metastasis often manifests within common tissues such as the brain, bone marrow, lung, and liver [12]. Based on this observation, Paget proposed the “seed and soil” hypothesis which posits that metastasis relies on the interaction and cooperation between cancer cells (seed) and a distant organ (soil). Consistent with this, cell-extrinsic features present within distant organs are fundamental in defining an organ’s receptiveness to metastatic colonization and outgrowth [8,13]. It is these extrinsic factors that support the capacity of DTCs to complete the metastatic cascade [5].

The formation of a ‘pro-metastatic niche’ in a distant organ is a fundamental determinant of DTC survival [5,14]. Changes in this niche environment can also signal for latent metastatic outgrowth by dormant cancer cells [15]. However, the mechanisms that direct the formation of a pro-metastatic niche can differ between distant organ sites. For example, the calprotectin heterodimer S100 calcium-binding protein A8 (S100a8) and S100 calcium-binding protein A9 (S100a9) regulates metastatic seeding and outgrowth in the lung [16] whereas SAA regulate the formation of a pro-metastatic niche in the liver [7,17]. Despite these differences in the initiating signals, the pro-metastatic microenvironment that forms ultimately converges on similar pro-inflammatory themes with the deposition of extracellular matrix (ECM) proteins and accumulation of myeloid cells [18]. ECM deposition within the pro-metastatic niche supports enhanced DTC migration across the endothelial layer and strengthens the anchoring of DTCs to the local parenchyma [19]. Further, the remodeling of the ECM can expose DTCs to trapped soluble factors necessary for DTC survival and proliferation. The ECM also supports the recruitment of bone-derived myeloid cells into the niche [20,21]. Notably, the polarization of these migratory myeloid cells within the niche helps to establish an immunosuppressive sanctuary and, in doing so, enables DTCs to evade immune elimination and ultimately heightens their metastatic potential [18,22]. Overall, the biological state of distant tissues is a critical determinant of the efficiency of the metastatic cascade.

## 3. The Acute Phase Response Supports Formation of a Pro-Metastatic Niche

Elements of the pro-metastatic niche bear striking resemblance to inflammatory processes that occur during an acute phase response (APR). The APR is a systemic nonspecific innate reaction that is evolutionarily conserved and occurs as a result of disturbances in homeostasis such as tissue damage [23,24]. Notably, cues that direct the formation of a pro-metastatic niche are often members of the acute phase protein (APP) family [23,25] (Figure 1). APPs are proteins primarily produced in the liver and by hepatocytes in response to pro-inflammatory cytokines, such as interleukin (IL)-6 and IL-1β, which are released in the setting of infection and tissue injury. However, this acute phase response requires a coordinated effort that involves not only hepatocytes, but also Kupffer cells, hepatic stellate cells, and liver endothelial cells. Together, APPs produced by these liver-resident cell populations act to maintain homeostasis and facilitate tissue repair [26,27,28]. The acute phase response also triggers recruitment of bone marrow-derived myeloid cells and fibrosis at the site of foreign invasion or tissue injury [25]. Among the acute phase reactants, proteins of the serum amyloid A family are the most prominent. There are four isoforms of SAA. Of these, SAA1 and SAA2 dominate and are released mainly by the liver. In humans, SAA3 is thought to be a pseudogene and has no clear biological role. In contrast, SAA4 is constitutively expressed by the liver and its expression is not altered by inflammatory stimuli [29,30,31]. Thus, SAA1 and SAA2, together referred to as SAA, are the main SAA isoforms in humans that are released during the APR.

In healthy individuals, circulating levels of SAA are detected at low levels. However, in response to stress or tissue damage, hepatocytes rapidly synthesize and release SAA. Hepatocytes are the main cellular source of SAA, although in obesity, adipocytes have also been shown to produce SAA [32,33,34]. SAA production is triggered by inflammatory molecules including IL-6, IL-1β, tumor necrosis factor (TNF), as well as microbial by-products (e.g., lipopolysaccharide, LPS), in a signal transducer and activator of transcription 3 (STAT3) dependent manner [35,36]. Within 4–5 h of triggering the APR, SAA are detected in the serum and eventually peak at serum concentrations reaching more than 1000-fold above baseline levels within 24 h [37,38]. This rapid and remarkable increase in SAA in the serum reflects the central role of SAA in liver biology and in regulating the acute phase response.

The biological effects of SAA are broad and seek to establish a balance between promoting and dampening inflammation [37]. Initially, SAA were described as an archetypal component of the acute phase response to infection where they are involved in the opsonization and elimination of invading pathogens [39]. For example, SAA are produced by the liver in response to respiratory and skin infections [40,41]. Upon binding to bacteria, SAA trigger the release of pro-inflammatory cytokines that recruit macrophages and neutrophils to the site of invasion. Infiltrating neutrophils then release reactive oxygen species (ROS) to kill bacteria, while recruited macrophages phagocytose the bacteria. The resolution of the infectious process causes SAA serum levels to return to baseline.

SAA mediates its effects by signaling through a range of distinct receptors, including toll like receptor (TLR)2, TLR4, formyl peptide receptor 2 (FPR2), scavenger receptor class B member 1 (SR-B1), receptor for advanced glycosylated end products (RAGE), and P2X purinoceptor 7 (P2X7R) (Figure 2) [37]. Structural studies show that SAA is non-pathogenic when it exists as a hexamer bound to high density lipoprotein (HDL) [42,43]. However, upon dissociation from HDL and aggregation into amyloid fibrils, SAA signals through its cognate receptors and induces the expression of anti-inflammatory cytokines that aim to resolve acute inflammation [29,44,45]. Notably, these receptors are expressed across a broad range of cell types derived from both hematopoietic and non-hematopoietic origins which illustrates the complexity of biological effects that SAA may produce. However, in contrast to the role of SAA in the acute setting, chronic conditions like inflammatory disorders, including rheumatoid arthritis, Crohn’s disease, and type 2 diabetes, lead to sustained and high serum levels of SAA [46,47]. Here, pro-inflammatory cues trigger the initial release of SAA but without a resolution of the inflammatory process, SAA production by the liver persists and supports a state of chronic inflammation. Similarly, this process can provoke an inflammatory response that inadvertently supports cancer development and metastasis [48,49]. To this end, formation of a pro-metastatic niche in the liver has been shown to be reliant on SAA [7]. Thus, cancer can co-opt acute phase biology to support metastasis.

## 4. SAA and Their Role in Supporting Liver Metastasis

SAA can act as a chemoattractant to facilitate tissue infiltration by monocytes and neutrophils [50]. Consistent with this, SAA produced in the setting of cancer facilitate the recruitment of myeloid cells to the liver. This cellular remodeling of the liver establishes a pro-metastatic niche environment supportive of the seeding and colonization of DTCs [7,13].

After lodging in the capillary bed of a distant organ, DTCs must anchor to the endothelial layer and extravasate into the tissue parenchyma to escape from circulation [51]. This is accomplished by mimicking leukocyte transendothelial migration, whereby circulating cells are captured by clusters of vascular cell adhesion molecules (V-CAM) and intercellular cell adhesion molecules (I-CAM) displayed by the cell junctions of activated endothelial cells present within the pro-metastatic niche [52,53]. To this end, SAA have the capacity to alter the liver vasculature to stimulate transendothelial migration. For example, SAA released from rheumatoid arthritis synoviocytes have been shown to interact with SR-B1 and FPR2 expressed on endothelial cells to induce the nuclear factor of kappa light polypeptide gene enhancer in B-cells inhibitor, alpha (IκBα) degradation, and upregulation of nuclear factor kappa-light-chain-enhancer of activated B cells (NF-κB) expression. Freed from IκBα, nuclear translocation of RELA (p65) then directs NF-κB signaling to increase the expression of ICAM-1 and VCAM-1 on hepatic vascular endothelium. Upregulation of these cell adhesion molecules facilitates monocyte and neutrophil adhesion to the vessel wall and their subsequent migration into the liver [54,55]. However, this activation of endothelium can also have deleterious effects. Specifically, SAA are central to the pathogenesis of uremia-induced atherosclerosis, a disease characterized by the stiffening and narrowing of the arteries [56]. The accumulation of free SAA protein on the arterial wall at the site of pathogenic lipoprotein invasion directs phagocyte migration by promoting expression of VCAM-1 on arterial endothelial cells [57,58]. Consequently, this process engenders the accumulation of foam cells and, thus, the formation of an atherosclerotic plaque [55,58]. Binding of SAA to RAGE within these atherosclerotic lesions then triggers the migration of vascular smooth muscle cells to further enable phagocyte transendothelial migration. This ultimately forms a pro-inflammatory feedback loop that manifests as the pathologic characteristics observed in atherosclerosis [56]. Similarly, SAA can promote myeloid cell transendothelial migration into the liver and in doing so, instruct the formation of a pro-metastatic niche which subsequently supports DTC extravasation into the liver [52,53].

SAA have been implicated in the deposition and remodeling of ECM proteins [7,18] which are key determinants of the migratory capacity of DTCs [7,18]. ECM remodeling in the pro-metastatic niche is supported by recruited neutrophils. In colorectal cancer, elevated levels of systemic tissue inhibitors of metalloproteinases (TIMP) metallopeptidase inhibitor 1 (TIMP-1) trigger hepatic stellate cells to release chemokines, such as C-X-C motif chemokine 12 (CXCL12), that recruit neutrophils to the liver [59]. Infiltrating neutrophils subsequently release extracellular traps (NETs) containing matrix metalloproteinases (MMPs) that cleave components of the extracellular matrix. In doing so, they release matrikines, peptides liberated by partial proteolysis of the ECM, that then signal for further recruitment of neutrophils to the liver and increased deposition of fibronectin and versican. This newly remodeled ECM then acts to provide additional anchors for DTC migration into the liver parenchyma [60,61,62]. Notably, this biology is analogous to the known role of SAA in directing ECM remodeling and enhanced leukocyte migration to sites of acute inflammation. For example, SAA can activate smooth muscle cells through TLR2 and in doing so, stimulate an increase in matrix metalloproteinase-9 (MMP9) expression, a known matrix metalloproteinase with the ability to degrade ECM proteins and to trigger further ECM remodeling through the recruitment of inflammatory myeloid cells [63,64]. Moreover, it has been shown that SAA and its attachment to the ECM can act as a scaffold that may support cell migration across the endothelial barrier. Specifically, CD4^+^ T-cells exhibit enhanced adhesion to immobilized SAA-ECM complexes in vitro [65,66]. Leukocyte binding to these complexes promotes increased TNF release, a cytokine known to be involved in endothelial cell modulation and the attraction of neutrophils and monocytes to sites of activation [67,68,69]. Taken together, these data suggest a central role for SAA in coordinating ECM remodeling which then supports the recruitment of inflammatory leukocytes and in doing so, establishes a pro-metastatic niche.

## 5. A Role for SAA in Coordinating the Inflammatory Response to Cancer

A hallmark of cancer is its ability to evade immune elimination. Immune escape is facilitated by multiple mechanisms including loss of tumor antigens, decreased cancer cell immunogenicity, and formation of an immune suppressive microenvironment [70]. In addition, systemic factors can influence the fitness of the immune system and in doing so, contribute to immune evasion [71,72,73]. In this regard, the liver is well-recognized for its capacity to promote immune tolerance [74]. Hepatic immune tolerance involves the triggering of T cell anergy, effector T cell elimination, the induction of regulatory T cells, and coordination of an immunosuppressive myeloid response [75,76,77]. Notably, the liver has been implicated in regulating immune surveillance in cancer [78]. This finding has spawned new research into understanding mechanisms by which the liver coordinates immune evasion.

The liver is a frequent site of metastasis and liver metastasis associates with decreased efficacy of immunotherapy [78,79,80]. This clinical observation suggests a role for the liver in regulating the efficacy of immunotherapy. Consistent with this, elevated acute phase proteins released by the liver associate with decreased efficacy of immunotherapy. For instance, a retrospective study of melanoma patients treated with immune checkpoint inhibitors targeting CTLA-4 and PD1 showed that decreased overall survival was associated with elevated baseline levels of IL6, high neutrophil-to-lymphocyte ratio, and elevated C-reactive protein (CRP) levels [81,82]. Elevated levels of CRP detected prior to treatment also associate with decreased progression-free and overall survival in patients with lung cancer receiving anti-PD1 therapy [83,84]. Similarly, a retrospective study found that baseline levels of SAA were prognostic for response to upfront anti-PD1 therapy in patients with advanced non-small cell lung cancer [85]. Overall, these retrospective studies suggest a direct biological influence of the liver on immune surveillance in cancer.

The release of SAA by the liver triggers hepatic inflammation that contributes to the formation of an immune sanctuary that supports the capacity of DTCs to evade immune elimination. Recently, the presence of cancer cells in the liver has been found to derail tumor immunity and the efficacy of immunotherapy in an antigen-specific manner [86]. Consistent with this, immature myeloid cells in the liver are known for their capacity to exert immune suppressive effects [87]. For example, monocyte-derived macrophages have been shown to trigger tumor antigen-specific T cell apoptosis in the liver via Fas/FasL interaction and in doing so, impair the efficacy of immunotherapy in mouse models of cancer [78]. In addition, arginase I released by neutrophils causes L-arginine catabolism leading to impaired T cell function [88]. The release of reactive oxygen species and nitrogen species by myeloid cells can also inhibit T cell functions by diminishing T cell proliferation and compromising T cell receptor activity [89]. Importantly, the immunosuppressive biology of myeloid cells is instructed by signals received from the surrounding microenvironment [90,91]. As such, myeloid cells may be fine-tuned by soluble factors released within the liver. For example, SAA have been shown to drive neutrophil differentiation by activating the TLR/myeloid differentiation primary response protein (MyD88) signaling pathway which then instructs neutrophils with a suppressive phenotype characterized by the expression of IL-10, an anti-inflammatory cytokine that is also involved in tissue repair [92,93]. Thus, SAA are not only fundamental to fueling the initial inflammatory response but also for shaping its phenotype.

The pleiotropic nature of SAA is informed, at least in part, by its unique structural properties. For example, SAA1 exists as a hexamer with binding sites for HDL and heparin. As a multimer, SAA1 can trigger the induction of pro-inflammatory cytokines, such as IL-6, TNF, and IL-1β. The multimerization and amyloid fibril formation of free SAA1 monomers is facilitated by key amino acid residues at the N and C terminals of the SAA1. Consequently, the removal of these residues ablates the ability of SAA1 to multimerize [94,95]. As a monomer, SAA1 lacks the capacity to interact with FPR2 to induce the release of pro-inflammatory cytokines by myeloid cells. Instead, myeloid cells respond to SAA1 monomers by activation of TLR2-dependent p38-MAP Kinase (MAPK) phosphorylation and release of IL-10 and C-C motif chemokine ligand 17 (CCL17). Together, these cytokines drive heightened arginase I activity and efferocytosis of apoptotic neutrophils [96]. For instance, mice co-treated with a SAA1 monomer and a lethal dose of LPS demonstrate a reduced expression of pro-inflammatory cytokines and improved survival [97]. Thus, the structure of SAA contributes to its ability to instruct the phenotype of an immune response.

Finally, the activation of the acute phase response may spoil immune surveillance in cancer. Immune activation against DTCs is reliant on “licensing” of dendritic cells (DC) which cross-present tumor antigens for priming of tumor-reactive CD8 T cells and natural killer (NK) cells [98,99]. However, inflammatory molecules, such as IL-6, derail DC generation and skew the differentiation of immature myeloid cells away from DCs and toward macrophages [100]. IL-6 also promotes a systemic and progressive deficiency in DCs in pancreatic carcinoma that is mediated, at least in part, by increased apoptosis [101]. In addition to the potential direct effects of IL-6 on DC biology, IL-6 also triggers the release of acute phase proteins by hepatocytes which may impinge on DC biology. For instance, CRP reduces the capacity of DCs to promote antigen-specific T cell expansion [102]. SAA have also been shown to inhibit DC differentiation via activation of TLR2 and FPR2 [103]. Here, SAA signaling in bone marrow cells results in decreased expression of transcription factor PU.1 (PU.1) and CCAAT/enhancer-binding protein alpha (C/EBPα), key transcription factors involved in DC differentiation [103,104,105]. TLR2 signaling in mature DCs also results in DC dysfunction in the setting of cancer [106]. Taken together, SAA and the acute phase response triggered by cancer may thwart the productivity of cancer immunosurveillance.

## 6. Determinants of Disseminated Tumor Cell Outgrowth

DTCs that successfully extravasate into a distant organ will either proliferate, die, or enter a state of dormancy. Cellular dormancy is an advantageous strategy utilized by DTCs to survive in a distant organ while adapting to the newfound microenvironment [107]. DTCs may lie dormant for decades before metastatic outgrowth is observed [15]. This clinical observation suggests the need for a secondary cue to trigger awakening and overt metastasis [108]. Consistent with this, local inflammation in the lung triggered by tobacco smoke has been shown to signal the recruitment of neutrophils and their release of neutrophil extracellular traps (NETs) which then support DTC awakening [109]. NETs are scaffolds of chromatin that contain proteases from the neutrophil’s secretory granules, including neutrophil elastase, cathepsin G, and MMP9 [110]. The release of NETs in tissues is intended to trap pathogenic invaders and mediate killing with the associated proteases. However, these proteases can also cleave the extracellular matrix that surrounds a dormant cancer cell. For instance, sequential cleavage of basement laminin by neutrophil elastase and then MMP9 reveals a neo-epitope that activates integrin signaling in dormant DTCs leading to their awakening [109]. This biology demonstrates the critical role of the surrounding local microenvironment in defining the fate of a DTC.

The precise role of SAA in regulating DTC dormancy is unknown. However, SAA have been shown to facilitate the formation of a microenvironment in the liver supportive of DTC proliferation [7]. In non-malignant settings, SAA trigger the inflammasome cascade by signaling through TLR2, TLR4, and P2X7R that are expressed by resident Kupffer cells. This signaling prompts assembly of the NOD-like receptor family pyrin domain containing 3 (NLRP3) inflammasome and release of IL-1β into the liver microenvironment. In doing so, IL-1β can reinforce SAA release by hepatocytes causing further activation of the inflammasome cascade in resident macrophages as well as monocytes and neutrophils that are subsequently recruited to the liver. As a result, a feed forward cycle is established [95,111]. As part of this cycle, SAA activates neutrophils via FPR2 to release IL-8 which signals for further neutrophil recruitment [112]. SAA signaling via TLR2 on migratory myeloid cells and granulocytes also results in granulocyte colony-stimulating factor (G-CSF) secretion, which amplifies the neutrophil response in the liver [113,114]. Overall, the liver inflammation that ensues in response to SAA release bears striking resemblance to the necessary elements needed to awaken DTCs.

## 7. SAA and Other Acute Phase Proteins Are Prognostic Tumor Biomarkers

Elevated acute phase proteins detected in the serum of patients with cancer are associated with poor prognosis. This relationship between inflammation and clinical outcome is described using the modified Glasgow Prognostic Score (mGPS), which combines the levels of acute phase proteins including CRP and albumin [48,115,116]. Elevated CRP (>10 mg/L) and decreased albumin levels (<35 g/L) correspond to a higher mGPS score that is tumor histology agnostic and correlates with cachexia, systemic inflammation, and poor outcome to cancer therapy [117]. In addition to CRP and albumin, hepcidin is an acute phase protein that is increased in the liver in response to IL-6 released in the setting of inflammation. Hepcidin is a peptide hormone involved in the regulation of iron metabolism [118]. Importantly, hepcidin suppresses the capacity of ferroportin to export iron for erythropoiesis and as a result, is implicated in the pathogenesis of anemia of chronic disease including anemia of cancer [119,120,121]. Elevated hepcidin levels have also been shown to correlate with poor prognosis for patients with pancreatic carcinoma, urothelial cancer, and renal cell carcinoma [122,123].

Like other acute phase proteins released by the liver, SAA have also been found to be prognostic across many solid cancers. For instance, elevated serum levels of SAA in patients with non-small cell lung cancer and colorectal cancer associate with liver metastasis [7]. SAA have also been identified as a biomarker for monitoring tumor relapse in nasopharyngeal cancer (NPC) [124]. Levels of SAA also correlate with outcomes in patients with advanced melanoma and when combined with CRP show prognostic potential for identifying patients with high-risk early-stage melanoma [125]. Further, SAA associate with poor outcomes to treatment in patients with newly diagnosed advanced pancreatic carcinoma [126]. Taken together, SAA and other acute phase proteins released by the liver have prognostic implications in cancer.

## 8. Future Considerations

The liver is vital to normal human physiology. It is responsible for supporting metabolism, immunity, digestion, and detoxification among many other functions. The liver is also a critical sensor of inflammation and coordinates an acute phase response aimed at restoring homeostasis. However, in the setting of cancer inflammation, this response by the liver results in pathology that can manifest in anemia, cachexia, immune dysfunction, and metastasis. As discussed in this Review, chronic release of acute phase proteins by the liver predicts treatment resistance and a poor prognosis. Mechanisms underlying the role of the liver in cancer pathogenesis are beginning to be understood. However, future studies will need to consider how to intervene on liver pathophysiology in the setting of cancer (Figure 3). One approach is to intervene on the triggers of liver inflammation. For example, blocking IL-6 signaling in hepatocytes has been shown to prevent the formation of a pro-metastatic niche in the liver and to restore the ketogenic response that is impaired in the setting of cachexia [7,127]. However, other factors including the gut microbiome [128] may also disrupt normal liver biology with clinical implications. To this end, it is possible that multiple determinants of liver inflammation will need to be controlled.

A second approach to intervening on liver biology is to provoke the resolution of inflammation in the liver. For example, macrophages are well recognized for their role in promoting liver fibrosis but are also essential for fibrosis resolution [129]. In this regard, myeloid activating agents (e.g., CD40 agonists) have been shown to induce macrophages with anti-fibrotic properties capable of remodeling the tumor microenvironment [130,131]. CD40 agonists are also known to induce liver inflammation [132,133], although their role in remodeling the pro-metastatic niche and liver fibrosis remains ill-defined. Alternatively, hepatocytes represent a novel therapeutic target based on their pivotal role in the production of acute phase proteins including SAA. In mouse models of cancer, STAT3 activation in hepatocytes is necessary for the acute phase response. Consistent with this, genetic deletion of STAT3 selectively in hepatocytes blocks the formation of the acute phase response and the subsequent formation of a pro-metastatic niche [7]. However, it is not known whether intervening on STAT3 once the acute phase response has initiated will trigger resolution and a return to normal liver homeostasis. Thus, while determinants involved in the resolution of liver pathology have been identified in non-malignant settings, the capacity to resolve liver pathology triggered by cancer is unknown.

A third approach is to block acute phase proteins and their interaction with downstream receptors involved in mediating pathology and amplifying the acute phase response. In support of this strategy, the genetic deletion of SAA has been shown to prevent formation of a pro-metastatic niche in the liver [7]. However, it remains unclear whether blocking the interaction of SAA with its cognate receptors will be effective in the setting of an ongoing acute phase response. Further, clinical grade reagents capable of inhibiting SAA are currently lacking. However, one strategy being studied is to use a 5-MER peptide (MTADV, methionine-threonine-alanine-aspartic acid-valine) derived from a pro-inflammatory CD44 variant. MTADV was found to bind to SAA in vitro and disrupt SAA aggregation as well as reduce colitis in an experimental model of inflammatory bowel disease [134]. Another approach is to intervene on SAA biology using antagonists that target SAA receptors. For example, SRB1 is a receptor for a SAA and a target for ITX 5061, which is an antiviral drug designed to inhibit SRB1 dependent hepatitis C virus entry into cells [135]. Similarly, SAA binds and activates FPR2 which is used by influenza to increase viral replication. Preclinical inhibitors of FPR2 have shown potential for blocking FPR2 [136]. SAA also interacts with RAGE and in doing so, triggers the secretion of tissue factor by monocytes which can be suppressed using RAGE antagonists [137]. Clinical grade inhibitors of RAGE are under evaluation in inflammatory diseases [138]. In addition, SAA activates the NLRP3 inflammasome via the P2X7 receptor leading to IL-1β secretion by macrophages [95]. Clinical grade P2X7 receptor antagonists have been evaluated as a treatment for Crohn’s Disease, rheumatoid arthritis, and neuroinflammatory disorders [139]. Finally, SAA bind to both TLR2 and TLR4 causing immune activation. Clinical grade inhibitors against both TLR2 and TLR4 have been developed [140]. Together, strategies to disrupt SAA biology may hold promise but the ability of SAA to signal through multiple receptors on many cell types represents a significant clinical challenge. To this end, a multi-faceted approach may be necessary wherein acute phase proteins, such as SAA, are targeted in concert with strategies that aim to intervene on the triggers of liver inflammation and to re-establish normal liver homeostasis.

Inflammatory networks induced by SAA and other acute phase proteins released by the liver may also have implications for patient stratification. Strategies to optimize patient selection of cancer therapies remains a major challenge [141]. Retrospective studies have shown that SAA and other acute phase proteins hold promise as prognostic biomarkers [124,125,126]. The acute phase response is a critical determinant of the cancer inflammation cycle and may act as rheostat to control immune homeostasis and risk for disease progression or relapse in patients [142]. Thus, we propose that monitoring the acute phase response in patients may ultimately prove to be informative for personalizing therapy to manage not only cancer but also its sequalae such as anemia and cachexia.

## 9. Conclusions and Outlook

SAA is a pleotropic cytokine, acting as a critical determinant of the biology of a wide range of human conditions including autoimmunity, infection, cardiovascular disease, and cancer. Here, we discussed an emerging role for SAA in cancer. SAA is released during an acute phase response and triggers inflammation aimed at re-establishing homeostasis. However, in the setting of cancer, the chronic release of SAA may undermine immune surveillance and foster the metastatic cascade thereby promoting disease progression. Consistent with this, SAA correlates with resistance to immunotherapy and poor prognosis overall. However, the precise mechanisms by which SAA coordinates liver metastasis and its impact on immune evasion in cancer remain to be elucidated. Nonetheless, the central role of SAA in instructing liver biology and promoting metastasis raises the possibility that it may be a therapeutic target. In this regard, the complex biology and structure of SAA poses many challenges. Notably, SAA signals through multiple receptors and can influence the biology of a wide range of cell types [143]. Future studies will need to address mechanisms leading to SAA release by the liver in the setting of cancer, strategies capable of intervening on SAA-directed biology, and the precise role of SAA in regulating the efficacy of cancer therapy. Together, this knowledge will guide the development of novel treatments focused on restoring liver homeostasis as an approach to improve patient outcomes in cancer.

## Figures and Tables

**Figure 1 cancers-13-03179-f001:**
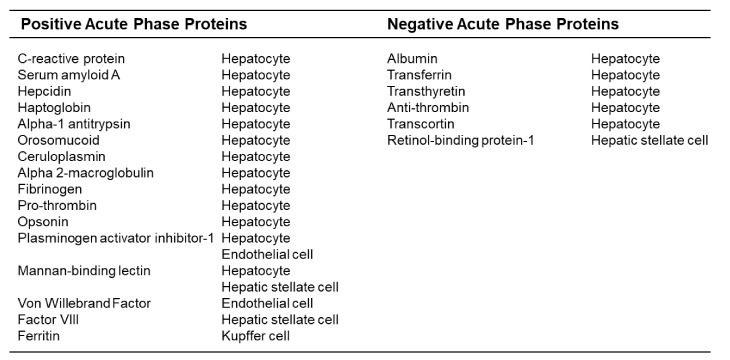
Liver-derived proteins associated with the acute phase response. Acute phase proteins are defined based on a change in plasma concentration of at least 25 percent in the setting of an inflammatory disorder. Shown are proteins that increase (positive acute phase protein) and decrease (negative acute phase protein) during an acute phase response. Also displayed is the predominant cellular source for each protein.

**Figure 2 cancers-13-03179-f002:**
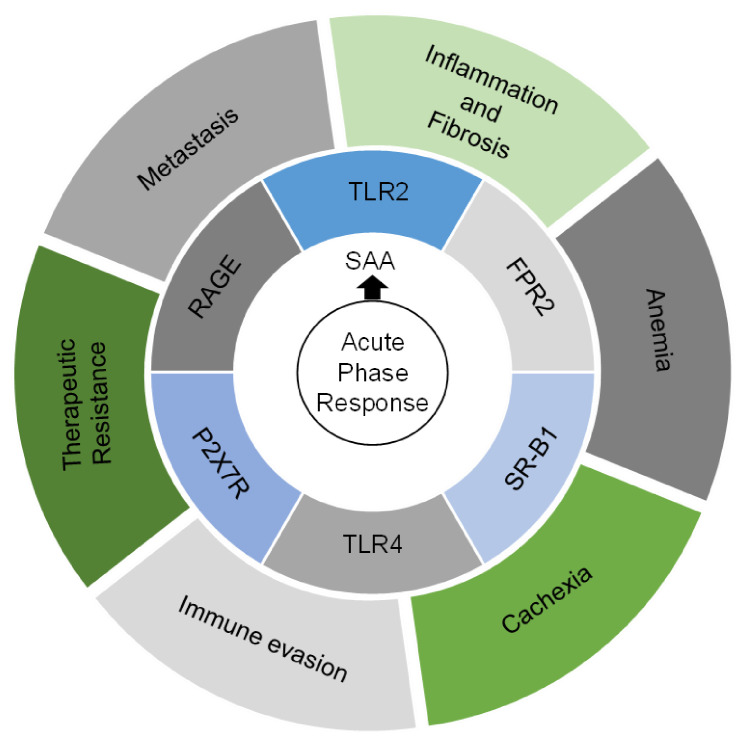
The acute phase response and its pathological impact in the setting cancer. This illustration encompasses six hallmarks associated with the acute phase response induced by cancer. Also shown are serum amyloid A (SAA) proteins which are released during the acute phase response and receptors that SAA proteins engage to support these hallmarks.

**Figure 3 cancers-13-03179-f003:**
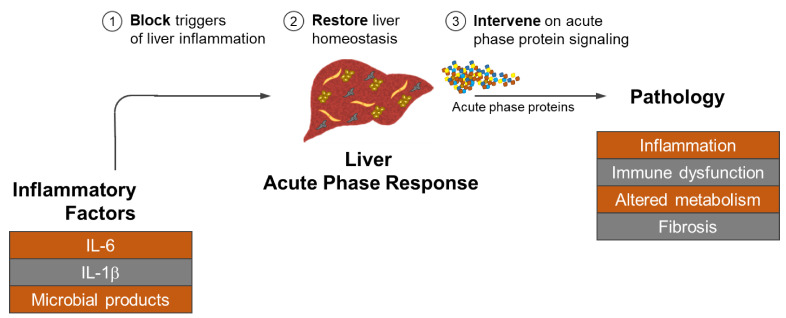
Strategies for intervening on the liver acute phase response in cancer. Inflammatory factors including IL-6, IL-1β, and microbial products (e.g., LPS) trigger the acute phase response in the liver leading to the release of acute phase proteins by activated hepatocytes, hepatic stellate cells, endothelial cells, and myeloid cells. Chronic release of acute phase proteins then mediate pathology by supporting inflammation, immune dysfunction (e.g., T cell apoptosis), alterations in metabolism (e.g., iron metabolism), and fibrosis. Shown are three potential approaches to intervene on the liver acute phase response by (1) blocking the activity of inflammatory factors that trigger liver inflammation (e.g., with blocking antibodies against IL-6), (2) restoring liver homeostasis by provoking a restorative immune response or disrupting the release of acute phase proteins by liver resident cells (e.g., via STAT3 inhibition), and (3) intervening on acute phase protein signaling by neutralizing acute phase proteins or inhibiting their interaction with downstream receptors (e.g., with TLR inhibitors).

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
