# Peer review of "Serum Amyloid A Proteins and Their Impact on Metastasis and Immune Biology in Cancer"

_cancers, 2021, doi:10.3390/cancers13133179_

Round 1

Reviewer 1 Report

The review manuscript by Lee and Beatty is a timely and extremely interesting prospective. It is well written and provides not only overview but interpretation and future considerations relevant to cancer therapeutics.

For an uninitiated audience it would be useful to have a definition of acute phase response. The clarity of what cells can secrete SAA was missing as there seems to be more than hepatocytes, and what is the secretion trigger? The authors state that there are other isoforms of SAA but do not further discuss the structural nature of this protein which would be helpful to appreciate the molecular aspects. How specifically does it bind its receptors? Or sufficient quantity of SAA triggers oligomerization and receptor aggregation? Perhaps a direction to other reviews would be helpful.

Several acronyms are not spelled out.

Author Response

Reviewer A: The review manuscript by Lee and Beatty is a timely and extremely interesting prospective. It is well written and provides not only overview but interpretation and future considerations relevant to cancer therapeutics.

Comment #1: For an uninitiated audience it would be useful to have a definition of acute phase response.

Reply: We appreciate the Reviewer’s suggestion and have revised the introduction to include a brief definition of the acute phase response. Specifically, we now state:

Line 74: “Elements of the pro-metastatic niche bear striking resemblance to inflammatory processes that occur during an acute phase response (APR). The APR is a systemic nonspecific innate reaction that is evolutionarily conserved and occurs as a result of disturbances in homeostasis such as tissue damage [23, 24].”

Comment #2: The clarity of what cells can secrete SAA was missing as there seems to be more than hepatocytes, and what is the secretion trigger?

Reply: As suggested by the Reviewer, we have revised our manuscript to include more detail on the cellular source of SAA and triggers of its release. Specifically, we now state:

Line 94: “In healthy individuals, circulating levels of SAA are detected at low levels. However, in response to stress or tissue damage, hepatocytes rapidly synthesize and release SAA. Hepatocytes are the main cellular source of SAA, although in obesity, adipocytes have also been shown to produce SAA [32-34]. SAA production is triggered by inflammatory molecules including IL-6, IL-1b, tumor necrosis factor (TNF), as well as microbial by-products (e.g. lipopolysaccharide, LPS) in a signal transducer and activator of transcription 3 (STAT3) dependent manner [35, 36]. Within 4-5 hours of triggering the APR, SAA are detected in the serum and eventually peak at serum concentrations reaching more than 1000-fold above baseline levels within 24 hours [37, 38].”

Comment #3: The authors state that there are other isoforms of SAA but do not further discuss the structural nature of this protein which would be helpful to appreciate the molecular aspects.

Reply:  As suggested by the Reviewer, we have now clarified the role of other isoforms and revised our manuscript as follows:

Line 88: “There are four isoforms of SAA. Of these, SAA1 and SAA2 dominate and are released mainly by the liver. In humans, SAA3 is thought to be a pseudogene and has no clear biological role. In contrast, SAA4 is constitutively expressed by the liver and its expression is not altered by inflammatory stimuli [29-31]. Thus, SAA1 and SAA2, together referred to as SAA, are the main SAA isoforms in humans that are released during the APR.”

Line 124: “Structural studies show that SAA is non-pathogenic when it exists as a hexamer bound to high density lipoprotein (HDL) [42, 43]. However, upon dissociation from HDL and aggregation into amyloid fibrils, SAA signals through its cognate receptors and induces the expression of anti-inflammatory cytokines that aim to resolve acute inflammation 29, 44, 45].

Line 250: “The pleiotropic nature of SAA is informed, at least in part, by its unique structural properties. For example, SAA1 exists as a hexamer with binding sites for HDL and heparin. As a multimer, SAA1 can trigger the induction of pro-inflammatory cytokines, such as IL-6, TNF, and IL-1b. Multimerization and amyloid fibril formation of free SAA1 monomers is facilitated by key amino acid residues at the N and C terminals of SAA1. Consequently, removal of these residues ablates the ability of SAA1 to multimerize [94, 95].”

Comment #4: How specifically does it bind its receptors? Or sufficient quantity of SAA triggers oligomerization and receptor aggregation? Perhaps a direction to other reviews would be helpful.

 Reply:  We thank the Reviewer for this suggestion. In our revised manuscript, we include references to Reviews that discuss the chemical and physical nature of SAA. Specifically, we now state:

Line 121: “SAA mediates its effects by signaling through a range of distinct receptors, including toll like receptor (TLR)2, TLR4, formyl peptide receptor 2 (FPR2), scavenger receptor class B member 1 (SR-B1), receptor for advanced glycosylated end products (RAGE), and P2X purinoceptor 7 (P2X7R) (Figure 2) [37]. Structural studies show that SAA is non-pathogenic when it exists as a hexamer bound to high density lipoprotein (HDL) [42, 43]. However, upon dissociation from HDL and aggregation into amyloid fibrils, SAA signals through its cognate receptors and induces the expression of anti-inflammatory cytokines that aim to resolve acute inflammation [29, 44, 45].”

Comment #5: Several acronyms are not spelled out.

Reply: We have made these corrections as suggested.

Reviewer 2 Report

The review written by Lee et al. is an excellent review that comprehensively describes the role of the SAA-coordinated early immune responses of the liver in the development of cancer. The followings are my comments.

 1.  Fig. 2 is the central figure showing the roles of SAA in cancer development. Here, the authors link the SAA receptors to their corresponding cancer hallmarks. However, the current figure lacks information on cell types responsible for these receptors and involve in these hallmarks. The information on the cell types will help readers understand the pivotal roles of SAA for these cancer hallmarks.

 2. Because the present review has no figure showing structural information on SAA itself. I think that the image of SAA will become clear if the authors give the structural information as a schema.  I believe the current Fig1 is a table. 

As a whole, the authors have organized the review well.

Author Response

Reviewer B: The review written by Lee et al. is an excellent review that comprehensively describes the role of the SAA-coordinated early immune responses of the liver in the development of cancer. The followings are my comments.

Comment #1: Fig. 2 is the central figure showing the roles of SAA in cancer development. Here, the authors link the SAA receptors to their corresponding cancer hallmarks. However, the current figure lacks information on cell types responsible for these receptors and involve in these hallmarks. The information on the cell types will help readers understand the pivotal roles of SAA for these cancer hallmarks.

Reply:  We appreciate the Reviewer’s comment. However, receptors for SAA are broadly expressed across a wide range of cells. For example, TLR2 is expressed not only by hematopoietic cells, including T cells, macrophages, dendritic cells, and B cells, but also many non-hematopoietic cell types, such as hepatocytes, endothelial cells, smooth muscle cells, and fibroblasts. Thus, the downstream signaling potential of SAA is quite complex. To clarify this, we have revised our manuscript as follows:

Line 128: “Notably, these receptors are expressed across a broad range of cell types derived from both hematopoietic and non-hematopoietic origins which illustrates the complexity of biological effects that SAA may produce.”

Comment #2: Because the present review has no figure showing structural information on SAA itself. I think that the image of SAA will become clear if the authors give the structural information as a schema. I believe the current Fig1 is a table. As a whole, the authors have organized the review well.

Reply: We appreciate the suggestion by the Reviewer to include structural information on SAA. In our revised manuscript, we have now expanded our discussion on the structure of SAA. In addition, we have included references to Reviews which provide more detail on the chemical and physical properties of SAA.

Line 121: “SAA mediates its effects by signaling through a range of distinct receptors, including toll like receptor (TLR)2, TLR4, formyl peptide receptor 2 (FPR2), scavenger receptor class B member 1 (SR-B1), receptor for advanced glycosylated end products (RAGE), and P2X purinoceptor 7 (P2X7R) (Figure 2) [37]. Structural studies show that SAA is non-pathogenic when it exists as a hexamer bound to high density lipoprotein (HDL) [42, 43]. However, upon dissociation from HDL and aggregation into amyloid fibrils, SAA signals through its cognate receptors and induces the expression of anti-inflammatory cytokines that aim to resolve acute inflammation [29, 44, 45].”

Reviewer 3 Report

Accept

Author Response

Reviewer C: No comments.

Reply: We thank the Referee for their review of our manuscript.